# Boar Seminal Microbiota in Relation to Sperm Quality under Tropical Environments

**DOI:** 10.3390/ani13243837

**Published:** 2023-12-13

**Authors:** CongBang Ngo, Junpen Suwimonteerabutr, Prasert Apiwatsiri, Imporn Saenkankam, Nuvee Prapasarakul, Jane M. Morrell, Padet Tummaruk

**Affiliations:** 1Department of Obstetrics, Gynaecology and Reproduction, Faculty of Veterinary Science, Chulalongkorn University, Bangkok 10330, Thailand; congbangdhnl39@gmail.com (C.N.); suwipen@hotmail.com (J.S.); 2Center of Excellent in Swine Reproduction, Chulalongkorn University, Bangkok 10330, Thailand; 3Department of Veterinary Microbiology, Faculty of Veterinary Science, Chulalongkorn University, Bangkok 10330, Thailand; prasert_315288@hotmail.com (P.A.); imporn.tfg@gmail.com (I.S.); nuvee.p@chula.ac.th (N.P.); 4Center of Excellence in Diagnosis and Monitoring for Animal Pathogens, Chulalongkorn University, Bangkok 10330, Thailand; 5Department of Clinical Sciences, Swedish University of Agricultural Sciences, 75007 Uppsala, Sweden; jane.morrell@slu.se

**Keywords:** bacteria, bioinformatics, pig, reproduction, semen

## Abstract

**Simple Summary:**

Semen possesses unique microbiota that are distinct from other microbial populations, such as those in the piglet gut and sow vagina. Environmental variations can result in alterations in the bacterial composition, which might not be detrimental but instead could be essential for sustaining optimal sperm functionality. The present study investigated the seminal microbiota in boars in relation to sperm quality. We found a negative linear relationship between the dominant bacterial orders, specifically potential probiotic bacteria (Lactobacillales) and harmful bacteria (Enterobacterales). Moreover, the abundance of potential probiotic functional bacteria was more frequently observed in high-quality ejaculates. Differences in the abundance of specific small subsets of microbes and their interactions revealed more precise effects on sperm quality than the overall seminal bacterial content (CFU/mL). The findings suggest that alterations in the seminal microbiome are significantly associated with the quality of boar semen, and strategies to increase the richness of functional candidates such as *Lactobacillus* spp. through herd management practices (e.g., seasonal changes, feeding, microbial exposure, antibiotics, and probiotics use) could create favorable reproductive potentials.

**Abstract:**

The present study was carried out to determine the seminal microbiota of boars and their correlation with sperm quality. A total of 17 ejaculates were collected from 17 Duroc boars and were classified according to sperm quality into two groups: low-quality (*n* = 8) and high-quality (*n* = 9). Each ejaculate was subjected to (i) semen evaluation, (ii) bacterial culture and MALDI-TOF identification, and (iii) 16S rRNA gene sequencing and bioinformatic analyses. No difference in the total bacterial count, alpha diversity, and beta diversity between the high-quality group and the low-quality group was detected (*p* > 0.05). While *Globicatella sanguinis* was negatively correlated with sperm quality (*p* < 0.05), *Delftia acidovorans* was positively correlated with sperm quality (*p* < 0.05). Lactobacillales (25.2%; LB) and Enterobacterales (10.3%; EB) were the most dominant bacteria and negatively correlated: EB = 507.3 − 0.5 × LB, R^2^ = 0.24, *p* < 0.001. Moreover, the abundance of *Escherichia-shigella* was negatively correlated with LB (r = −0.754, *p* < 0.001) and positively correlated with *Proteus* (r = 0.533, *p* < 0.05). *Alysiella* was positively correlated with *Lactobacillus* (r = 0.485, *p* < 0.05), *Prevotella* (r = 0.622, *p* < 0.01), and *Staphylococcus* (r = 0.489, *p* < 0.05). In conclusion, seminal microbiota is significantly associated with boar semen qualities. The distributions of the most dominant bacterial genera, the differences in the abundance of small subset microbes, and their correlation appear to have far more impact than the overall seminal bacterial content (e.g., total bacterial count, alpha diversity, and beta diversity) on sperm quality.

## 1. Introduction

Boar semen quality and infertility have frequently been linked to bacterial contamination [1,2,3]. However, the precise effects of bacteria on semen quality and the overall microbiota of boar seminal fluid have not been fully elucidated [4,5]. It appears that semen has distinct microbiota when compared to other microbiome populations, such as the piglet gut and sow vagina, and it can be hypothesized that these microbiota compositions may not be harmful but rather necessary for maintaining optimal sperm function [6,7]. The host immune system is positively modulated and balanced by the intestinal microbiome, which has been shown to improve gut health, prevent infection, and reduce antibiotic use [8,9,10]. This new information opens a remarkable and developing field of research that is advancing our knowledge of the etiology of both male and female fertility, by examining the seminal and vaginal microbiome [11,12]. However, little research has been conducted on the seminal microbiome in boars [5,7] and no studies have been conducted in tropical environments. Moreover, a comprehensive study on the association between the boar seminal microbiome and sperm quality has not been performed in pigs. 

Based on microscopic-culture-dependent methods and the polymerase chain reaction (PCR) technique, earlier studies focused mainly on the detection of identified pathogens [13,14,15]. However, these techniques underestimated the abundance of dominant, small-subset microbes and their correlations in boar semen. During the last decade, next-generation sequencing and bioinformatics analyses have made it feasible to determine the bacterial composition of semen more accurately [5,7]. Furthermore, this technique also reveals more about the interaction between the host and the bacterial community via parameters such as the microbial richness and diversity (alpha diversity), the shifts in overall microbiota composition between two different communities (beta diversity), and the bacteria abundance, building up a classified taxonomic profile (heat map) [7,11,12].

The influence of genetics, such as boar breeds; environmental factors, including the variations in environments, microbial exposure, season, feeding, use of antibiotics and probiotics, and quality of hygienic measures; and the process of semen collection all contribute to differences in microbial richness and diversity in bacterial composition that are related to the host immune system [5,11,16,17]. Under the Mediterranean climate of Catalonia, Spain, three dominant species within the Pietrain seminal microbiome are *Bacillus megaterium*, *Brachybacterium faecicum*, and *Bacillus coagulans* [4]. In contrast, in the humid subtropical climate of Shanghai, China, *Pseudomonas* and *Lactobacillus* are the dominant genera in ejaculates collected from Duroc, Landrace, and Yorkshire boars [5]. These variations among different boar semen microbiome populations indicate that investigating seminal microbiomes under specific conditions can reveal new insights into the etiology of male fertility.

During summer, the seminal bacterial richness and diversity in Landrace boars are higher than in Duroc and Yorkshire [5]. *Pseudomonas* is more common in the summer and negatively correlates with sperm quality and reproductive performance, whereas *Lactobacillus* is more prevalent in the winter and is favorably associated with sperm quality and reproductive potential [5]. Consequently, the increase in supplemental vitamins and organic antioxidants during the summer months at commercial artificial insemination boar studs has been suggested [18,19]. However, differences in the boar seminal microbiota in relation to various semen qualities have not been comprehensively investigated in tropical environments. The current research was conducted to identify the composition of the seminal microbiota in boars at a Thai breeding facility and to explore its correlation with the quality of sperm. 

## 2. Materials and Methods

### 2.1. Animals

The semen samples used in this study were obtained from boars at a breeding facility in the western region of Thailand. The experiment was conducted from November to December 2021. There were 17 Duroc boars, each of which produced one ejaculate. All of the boars had previously been proven fertile and had an average age of 1.91 ± 0.46 years (ranging from 1.1 to 2.8 years) and an average body weight of 262.4 ± 37.7 kg (ranging from 193 to 342 kg). The boars were nourished in a closed structure with an evaporative cooling system, in separate pens (9 m^2^/boar). The barn had an average temperature and humidity of 22.5 ± 0.7 °C and 72.0 ± 1.2%, respectively. Each boar received a typical daily diet of 2.5–3.2 kg of commercial feed and unlimited access to water via water nipples.

### 2.2. Experimental Design

All the semen samples were evaluated both macroscopically and microscopically, including semen volume (mL), sperm concentration (×10^6^ sperm per mL), the total number of sperm per ejaculate (×10^9^ sperm), pH, sperm motility (%), sperm viability, sperm motility (%), acrosome integrity (%), sperm membrane functionality (%), and mitochondrial activity (%). In addition, 0.5 mL of each ejaculate was used for the bacterial culture aerobically, evaluating the total bacterial count (CFU/mL, log_10_) and identifying isolated bacteria using matrix-assisted laser desorption ionization–time-of-flight mass spectrometry (MALDI–TOF MS, Microflex^®^ LT, MALDI Biotyper™ System, Bruker, Bremen, Germany). Moreover, the ejaculates were also subjected to PCR amplification to determine the microbiota using 16S rRNA gene sequencing and a bioinformatics analysis, including alpha diversity, beta diversity, and abundance testing. Correlation analyses were conducted to examine the association between semen characteristics, total bacterial count, and the concentration of major identified bacteria. Furthermore, sperm viability was assessed on a scale from 1 to 17, with 1 being the lowest and 17 the highest, across the 17 collected ejaculates. This evaluation facilitated the categorization of the samples into two groups based on semen quality: a low-quality group with an average viability of 71.3 ± 1.3% (*n =* 8), and a high-quality group with an average viability of 83.6 ± 1.2% (*n =* 9). In addition, the study measured other sperm quality parameters such as motility, acrosome integrity, membrane integrity, and mitochondrial activity, comparing these between the low- and high-quality semen groups (Table 1). Finally, the study compared the total bacterial count and the concentrations of major identified bacteria using different techniques between the low- and high-quality semen samples.

### 2.3. Semen Collection and Evaluation

The semen was collected using the gloved-hand method. Immediately after collection, the semen volume and pH were measured. Sperm concentration was evaluated by Spermacue^®^ (Minitube, Tiefenbach, Germany). To calculate the total sperm per ejaculate, semen volume was multiplied with sperm concentration [20,21]. Various parameters including total sperm motility, progressive motility, straight-line velocity (VSL, µm/s), curvilinear velocity (VCL, µm/s), average path velocity (VAP, µm/s), straightness (STR, %), linearity (LIN, %), and wobbe (WOB, %) were all assessed using a computer-assisted sperm analysis system (SCA^®^ CASA System, MICROPTIC S.L., Barcelona, Spain). The CASA system was configured for boar sperm with a frame rate of 50 frames per second and a box size of 100 pixels. The objects had minimum and maximum areas ranging from 10 µm^2^ to 80 µm^2^, respectively. The spermatozoa with motility were configured for static (10 µm/s), slow–medium (25 µm/s), and progressive motility (>45 µm/s). The semen was diluted with phosphate-buffered saline (PBS; 1:10) and then put in Hamilton 2X-CEL^®^ Slides, Disposable Sperm Analysis Chamber, 20 microns (Hamilton Thorne Inc, Massachusetts, USA), and analyzed using a phase-contrast microscope at 37 °C on a warmed stage (TOKAI HIT, Shizuoka-ken, Japan) (BX41, Olympus, Shinjuku, Japan). Phosphate-buffered saline, with a pH of 7.2–7.4, was used to protect sperm cells from osmotic damage, such as rupturing or shrinking [22]. In total, 1500 spermatozoa in five randomly chosen fields from each sample were used to calculate the proportion of motile sperm [3]. 

The assessment of sperm viability was conducted using SYBR-14/EthD-1 (Fertilight^®^ Sperm Viability Kit, Molecular Probes Europe, Leiden, The Netherlands). In this procedure, a 10 µL portion of diluted semen was combined with 1 µL of 14 µM EthD-1 (Molecular Probes Inc., Eugene, OR, USA) in 1 mL of PBS. Additionally, 2.7 µL of 0.38 µM SYBR-14 (Dead/Alive Kit; Molecular Probes Inc.) in 1 mL of dimethyl sulfoxide (DMSO) was added before the mixture was incubated at 37 °C for 15 min. Subsequently, examination of 200 sperm cells was carried out using a fluorescence microscope (1000×; CX-31; Olympus, Tokyo, Japan). Sperm cells exhibiting only a green stain were classified as alive with an intact plasma membrane. Those stained red or both red and green were categorized as dead or having damaged plasma membranes, respectively. Sperm viability was determined based on the proportion of living sperm with an intact plasma membrane [3,23].

For assessing acrosome integrity, EthD-1 (Fertilight^®^, Sperm Viability Kit, Molecular Probes Europe, Leiden, The Netherlands) and fluorescein isothiocyanate-labeled peanut (Arachis hypogaea) agglutinin (FITC-PNA) staining (Sigma-Aldrich Co., Ltd., St Louis, MO, USA) were used. In this process, a 10 µL aliquot of diluted sperm was combined with a 10 µL aliquot of 14 µM EthD-1 (Molecular Probes Inc., Eugene, OR, USA) and incubated at 37 °C for 15 min. Following incubation, an 8 µL drop of the sperm sample was spread on a slide, air-dried at room temperature, and then dipped for 30 s in 95% ethanol. After staining with 15 µL of the FITC-PNA solution (FITC-PNA in PBS (1:10, *v*/*v*)) at 4 °C for 30 min in a moist environment, the slide was washed with a cold PBS solution. Finally, using fluorescence microscopy (1000×; CX-31; Olympus, Tokyo, Japan), the status of 200 sperm cells per sample was examined. Sperm cells with an orange acrosome cap, a green band at the equatorial segment, or the acrosome cap disrupted into a patch-like pattern were all classified as negative. The fraction of spermatozoa stained green (positive) with intact acrosome caps was determined [23].

To evaluate sperm membrane functionality, the short hypo-osmotic swelling test (sHOST) was used. In a 1.5 mL Eppendorf tube, a 10 µL aliquot of a diluted sperm sample and 200 µL of a citrate buffer (75 mOsM) were combined and incubated in the dark for 30 min at 37 °C. Following incubation, 175 µL of a HOS solution containing 5% formaldehyde (75 mOsM) was added. Finally, using light microscopy (400×; CX-31; Olympus, Tokyo, Japan), an 8 µL drop of the sample was deposited on a glass slide to analyze the sperm plasma membrane permeability of 200 sperm cells per sample. Sperm cells were classed as either positive (having a coiled tail) or negative (having a straight tail). The fraction of positive sperm suggests that the sperm membrane is functioning [23]. 

To assess sperm mitochondrial activity, fluorochrome 5,5′,6,6′-tetrachloro-1,1′,3,3′-tetraethylbenzimidazoly-carbocyanine iodide (JC-1; Molecular Probes, Molecular Probes Inc., Eugene, OR, USA) was used. In this process, 25 µL of the JC-1 solution composed of 1.6 µL of 0.153 mM JC-1, 1 µL of 0.02 mM SYBR-14, and 1.6 µL of 2.4 mM PI in 100 µL of a HEPES-buffered medium was mixed with 12.5 µL diluted semen, then incubated at 37 °C for 30 min. Finally, using a fluorescent microscope (1000×; CX-31; Olympus, Tokyo, Japan), one drop of 8 µL of the dyed sperm sample was placed on a glass slide to analyze the mitochondrial activity of 200 sperm cells. Spermatozoa with a high mitochondrial membrane potential (positive) exhibited yellow-orange fluorescence at the midpiece, whereas spermatozoa with a low mitochondrial membrane potential (negative) exhibited little or no green fluorescence [3,23]. 

### 2.4. Bacterial Culture, Identification, and Quantification

One milliliter of the raw semen samples was diluted in tubes containing 9.0 mL of PBS (0.1 M phosphate buffer containing 0.15 M NaCl, pH 7.3) to prepare for serial dilution (10^0^–10^3^). From each dilution, 1.5 mL was placed on three different count agar plates (0.5 mL/plate) and incubated aerobically at 37 °C for 24 h. Concentration was expressed in colony forming units per milliliter (CFU/mL). The counting range was 30 ≤ X ≤ 300, in which plates with less than 30 CFU/mL or more than 300 CFU/mL were classified as too few to count (TFTC) and too many to count (TMTC), respectively. The average number of colonies on three different plates was used to calculate the total number of bacteria (CFU/mL). The total number of aerobic bacteria was log_10_ transformed and used in statistical analyses [24,25]. The colonies that were visible after 24 h of aerobic incubation at 37 °C were selected for identification using matrix-assisted laser desorption ionization–time-of-flight mass spectrometry (MALDI-TOF MS) (microflex^®^ LT, MALDI Biotyper™ System, Bruker, Bremen, Germany). Bacteria were identified by comparing the unknown organism’s peptide mass fingerprint (PMF) to the PMFs in the database [26,27]. Colonies were selected from blood agar plates and then analyzed in a Bruker Biotyper MALDI-TOF with 1 µL of formic acid (FA) and 1 µL of an alpha hydroxyl 4 cinnamic acid matrix (HCCA) [28].

### 2.5. DNA Extraction, PCR Amplification, and Bioinformatics Analysis

The DNA was extracted from 3 mL of raw semen using a QIAamp^®^ DNA mini kit (Qiagen AG, Basel, Switzerland). The process was started by separating sperm and seminal plasma by centrifugation at 800× *g* for 10 min and then the seminal plasma was centrifuged at 5000× *g* for 20 min to pellet the bacteria. Thereafter, the bacterial pellet was suspended in 200 µL of a lysis buffer (20 mg/mL of lysozyme; 20 mM Tris·HCl, pH 8.0; 2 mM EDTA; 1.2% Triton, 43 mM DTT) and incubated for 30 min at 37 °C before adding proteinase K and buffers following the manufacturer’s instructions (Baud et al., 2019 [7]). Finally, the DNA was eluted in 60 µL of buffer AE to increase the final concentration. The DNA concentration and purity were measured using a NanoDrop microvolume spectrophotometer (OneDrop TOUCH Lite, Biometrics Technologies Inc., Nonthaburi, Thailand). The DNA samples that met the requirements of (i) a DNA concentration of 15 ng/µL, (ii) a purity ratio of 260/280 > 1.8, and (iii) a purity ratio of 260/230 > 1.5 were used for further analyses. 

Bacterial 16S rRNA was amplified using a pair of universal primers targeting the V3–V4 regions and 2 × sparQ HiFi PCR master Mix (QuantaBio, MA, USA). The pair of a 16S amplicon PCR forward primer and PCR reverse primer combined the overhang adapter sequences (the underlined sequence) and the 16S V3V4 region (the rest sequence) [29]. Forward: 5′CGTCGGCAGCGTCAGATGTGTATAAGAGACAGCCTACGGGNGGCWGCAG and Reverse: 5′GTCTCGTGGGCTCGGAGATGTGTATAAGAGACAGGACTACHVGGGTATCTAATCC.

The amplification condition was started with the initial denaturation step at 98 °C for 2 min, followed by 30 cycles of 98 °C for 20 s, 55 °C for 30 s, and 72 °C for 60 s, and completed by a step of final extension at 72 °C for 1 min. Then, 16S amplicons were purified using sparQ Puremag Beads (QuantaBio, Beverly, MA, USA) and indexed using 2.5 µL of each Nextera XT index primer in a 50 µL PCR reaction, followed by eight cycles of the PCR condition above. Cleaning, pooling, and diluting the final PCR products to a final loading concentration of 4 pM were performed. On an Illumina MiSeq device, cluster formation and 250 bp paired-end read sequencing were carried out. 

A bioinformatic analysis of the microbiome was performed with QIIME 2 version 2020.8 [30]. Through next-generation gene sequencing, we extracted and sequenced DNA to identify total bacterial flora contaminated in the boar semen. We interpret the total isolated bacterial flora by examining (i) dominant abundant bacteria, a small subset of bacteria and their correlations, and (ii) the bacterial richness and diversity in each semen sample (alpha diversity: Chao-1, Shannon index), as well as among semen samples (beta diversity: UniFrac distance). In order to obtain pure data for data construction purposes, raw sequence data were demultiplexed, then quality filtered using the q2-demux plugin and finally denoised via q2-DADA 2 [31]. SATé-enabled phylogenetic placement (SEPP)–Greengenes 13.8 sequences were applied to construct the phylogenetic tree [32]. Alpha diversity was estimated via observed OTUs. The Chao1 richness and Shannon index were calculated separately for each sample and for the two experimental groups [33]. The phylogeny-based weighted beta diversity UniFrac (presence–absence phylogenetic distance) was applied to calculate the overall microbial community composition [34,35]. After samples were rarefied to 1611 sequences per sample, UniFrac distance metrics, which represent the characteristics between microbial communities of the sample, were transformed into principal coordinates using a principal coordinate analysis (PCoA) to visualize the sample distribution patterns [36]. Taxonomy was assigned to amplicon sequence variants (ASVs) using the q2-feature-classifier, which is the classify-sklearn naive Bayes taxonomy classifier trained on the Silva 138 99% OTU reference sequences [37].

### 2.6. Statistical Analysis

The data were analyzed using SAS software 9.4 (SAS Inst. Inc., Cary, NC, USA). For testing normal distribution of the continuous data, the Shapiro–Wilk test and Qualitative–Quantitative plots (Q–Q plots) were used. Semen characteristics and bacteria contamination between low- and high-quality ejaculates were analyzed using the general linear model procedure (GLM) of SAS (Table 1). Variables describing various traits of sperm production and contaminants, i.e., semen volume, sperm concentration, total number of sperm per ejaculate, pH, sperm motility and motion characteristics (VSL, VCL, VAP), sperm viability, acrosome integrity, sperm membrane functionality and mitochondrial activity, and total bacteria count (CFU/mL log_10_), were regarded as dependent variables and included in the statistical models. The statistical models included the semen quality group (low-quality and high-quality) as the main effect. Least-squares means were obtained from each class of factors and compared using the least significant difference test. Statistical analyses for true diversity and richness using the Shannon index, Observed Features, and Chao1 (alpha diversity) were conducted to compare the low- and high-quality groups using the Mann–Whitney U test. Spearman correlation was performed to analyze the correlations among the major identified bacteria and semen characteristics. The shifts in overall microbiota composition (beta diversity) were calculated using Pseudo F. The linear relationship between dominant abundance Lactobacillales (LB) and Enterobacterales (EB) was analyzed using regression analyses. Differences with *p* < 0.05 were considered to be significant.

## 3. Results

### 3.1. Semen Characteristics and Total Bacterial Count across Semen Samples

On average, sperm viability, sperm motility, sperm membrane functionality, and mitochondrial activity in the high-quality group were higher than in the low-quality group (*p* < 0.05) (Table 1). However, no difference in the total bacterial count between the high-quality group and the low-quality group was detected (*p* > 0.05) (Table 1).

### 3.2. The Richness and Diversity across Semen Samples

Regarding alpha diversity, although there was a higher tendency of a bacteria feature observed in low-quality samples compared to high−quality ones (10.0 ± 10.4 versus 8.1 ± 10.4, respectively, *p* > 0.05), there were no differences in the Chao1 and Shannon index between the low- and high-semen-quality groups, 8.3 ± 10.4 versus 9.8 ± 10.4 and 8.6 ± 10.4 versus 9.5 ± 10.4, respectively, *p* > 0.05). There was no difference in the shifts of overall microbiota composition (beta diversity) between boar semen of low and high quality (*p* = 0.433) (Figure 1a). In addition, the rarefaction curve represents a tendency of a gradual increase in the bacteria diversity across ejaculates expressed via the Shannon index in which the highest index was 6.85 and the lowest was 2.16 (Figure 1b). 

### 3.3. Bacteria Identification Using Bacterial Culture and Next-Generation Sequencing Methods

The bacteria detected in all ejaculates, using both a bacterial culture and next-generation gene sequencing methods, are described in Figure 2. Differences regarding dominant microbes between the two methods were observed. While *Staphylococcus* spp., *Micrococcus* spp., *Globicatella sanguinis*, *Proteus*, and *Pseudomonas aeruginosa* accounted for the highest proportion of microbes isolated using the bacterial culture method (Figure 2a), microbes belonging to the Lactobacillales order, Bacilli class, *Escherichia–Shigella*, and *Porphyromonas* were dominant bacteria detected by gene sequencing (Figure 2b). Using the conventional method for bacterial identification, *Staphylococcus* spp. accounted for the highest proportion of bacteria in both low- and high-quality groups, with 8.9% and 30.5%, respectively. However, when evaluated using the sequencing technique, Lactobacillales accounted for the highest abundance in both low- and high-quality groups with 14.1% and 9.7%, respectively.

The correlations among major culture-identified bacteria and semen characteristics are presented in Table 2. *Staphylococcus* spp., *Corynebacterium* spp., and *Escherichia coli* were positively correlated with the pH of semen (*p* < 0.05). Significant negative correlations were detected between *Staphylococcus* spp. and sperm membrane functionality (r = −0.577, *p* < 0.05), *Micrococcus* spp. and sperm membrane functionality (r = −0.529, *p* > 0.05), and *Proteus* spp. and acrosome integrity (r = −0.581, *p* < 0.05). *Globicatella sanguinis* was negatively correlated with the semen characteristics (*p* < 0.05), while *Delftia acidovorans* was positively correlated with them (*p* < 0.05) (Table 2).

### 3.4. Boar Seminal Microbiota Profile in Relation to Semen Quality

The microbial profile of boar semen samples is depicted in Figure 3. Firmicutes (49.5%), Proteobacteria (22.6%), Actinobacteriota (12.5%), and Bacteroidota (12.4%) emerged as the predominant phyla across all samples. The most dominant bacterial orders were Lactobacillales (LB), Enterobacterales (EB), and Bacteroidales, accounting for 25.2%, 10.3%, and 9.3%, respectively (Figure 3). The percentage of Lactobacillales varied widely, ranging from 2.2% to 71.9% across the 17 samples. When these percentages were grouped into quartiles, the first quartile (Q1)—which had the lowest percentage of LB—mostly comprised low-quality semen samples. The second (Q2) and fourth quartiles (Q4) included a mix of both high- and low-quality samples, while the third quartile (Q3) exclusively contained high-quality semen samples (Figure 4). Significant differences in the abundance of certain bacteria, such as *Alysiella*, *Myroides*, and Bacilli, were noted between the low- and high-quality ejaculates (Table 3). However, no significant differences were observed in other genera, including *Acinetobacter*, *Campylobacter*, *Proteus*, and *Staphylococcus* (Table 3).

### 3.5. Correlation between Major Bacteria Detected by Next-Generation Sequencing Method

A negative linear relationship was detected between the abundance of Lactobacillales (LB) and Enterobacterales (EB) in which the EB was decreased by 0.5 for every unit increase in the LB: EB = 507.3 − 0.5 × LB, R^2^ = 0.24, *p* < 0.001 (Figure 5). Correlations between major bacteria detected by next-generation sequencing are presented in Table 4. Interestingly, *Escherichia–shigella* was negatively correlated with Lactobacillales (r = −0.754, *p* < 0.001) but positively with *Proteus* (r = 0.533, *p* < 0.05). Moreover, *Alysiella* was positively correlated with *Campylobacter* (r = 0.714, *p* < 0.01), *Corynebacterium* (r = 0.723, *p* < 0.01), *Lactobacillus* (r = 0.485, *p* < 0.05), *Prevotella* (r = 0.622, *p* < 0.01), *Staphylococcus* (r = 0.489, *p* < 0.05), and *Streptococcus* (r = 0.547, *p* < 0.05). In addition, *Myroides* was negatively correlated with *Aerococcaceae* (r = −0.614, *p* < 0.01), *Campylobacter* (r = −0.486, *p* < 0.05), and *Globicatella* (r = −0.503, *p* < 0.05) (Table 4).

## 4. Discussion

### 4.1. Boar Seminal Microbiota 

Across all the detected phyla, Firmicutes (49.5%), Proteobacteria (22.6%), Actinobacteriota (12.5%), and Bacteroidota (12.4%) were the dominant phyla. The bacterial phyla profile in the present study was in agreement with Zhang et al. [5], who found that Proteobacteria (57.5%), Firmicutes (31.2%), Bacteroidetes (4.2%), and Actinobacteria (3.4%) were the most abundant phyla in boar semen. Similarly, Gòdia et al. [4] revealed that after sequencing 40 Pietrain ejaculates collected in Spain, the dominant phyla were Proteobacteria (39.1%), Firmicutes (27.5%), Actinobacteriota (14.9%), and Bacteroidota (5.7%). The consistency among seminal bacterial communities reported in previous studies indicates the comparability and authenticity of our sequencing data. It appears that semen has its specific microbiota, which contain bacteria that are beneficial or harmful to sperm quality, and the presence of a specific organism may not be deleterious but rather necessary to create a balanced or even improved sperm function [6,11]. 

In the present study, bacteria in the boar semen samples were subjected to 16S rRNA gene sequencing, and the microbiome profiles among samples were constructed based on a comparison with an equal number of 1664 sequences to minimize the sequence artefact resulting from high-throughput sequencing [38]. Our study indicates that a total of 3,609,615 sequence reads were obtained in which there were 17 phyla and 270 genera present in the 17 semen samples. These numbers are lower than a study conducted in China, which detected up to 5,450,539 sequence reads, 24 phyla, and 291 genera [5]. This difference can be explained by the higher number of boar semen samples that Zhang et al. [5] used for sequencing: 120 samples compared to 17 in our study. Also, the wholeness of the boar seminal microbiota is more elaborated when looking at the rarefaction curve created in the current study. A tendency for a continuous increase in the Shannon diversity index indicated that the 17 samples used in the present study contribute significantly to the boar seminal microbiota but were not deep enough to reveal the whole seminal bacteria communities. The latter not only requires a large number of samples for sequencing but will also depend on specific endemic environments [16,39].

### 4.2. Boar Seminal Microbial Richness and Diversity 

It has been hypothesized that semen quality is associated with the distribution of dominant bacterial genera present. This hypothesis was confirmed in the present study, which identified a negative linear relationship between the abundance of LB and EB. Moreover, when visualizing the distribution of bacterial communities of 17 samples using PCoA, classified by an increasing percentage of LB, cluster Q1 predominantly contained low-quality semen samples. In contrast, clusters Q2 and Q4 comprised a mixture of both high- and low-quality samples, whereas cluster Q3 exclusively contained high-quality semen samples. This finding suggests that the abundance of potentially probiotic functional bacteria is more frequently observed in high-quality ejaculates. Weng et al. [11] also corroborated the hypothesis that seminal bacteria communities are highly associated with semen quality in humans. Similarly, Zhang et al. [5] reported a strong association between seminal bacterial communities and both semen quality and fertility in boars.

In this investigation, no differences were observed in the microbial richness or diversity (alpha diversity) nor in the overall changes in microbiota composition (beta diversity) between ejaculates of low and high quality. This outcome aligns with the findings of Baud et al. [7], who also reported no variance in the estimated number of species, their distribution evenness, or microbiota composition shifts across samples. However, these results stand in contrast to those reported by Zhang et al. [5], who observed a higher alpha diversity in boar seminal bacteria in winter samples as opposed to summer ones. Additionally, in the summer, Landrace boars exhibited higher Chao1 and Shannon indices compared to Duroc and Yorkshire breeds. This could be explained through the differences in genetic capacity among boar breeds to adapt to changes in the environment, especially when heat stress is recognized as the main factor in summer infertility [14,19]. The lack of significant differences in alpha and beta diversity in this study could be attributed to factors such as seasonal changes and the breed of boar, considering all samples were from Duroc boars and collected during the cooler season. Kraemer et al. [39] noted that during the summer, UV radiation exposure can reduce bacterial levels in boar pens. Moreover, the increased use of antibiotics as a disease prevention strategy during the summer could further contribute to a reduction in bacterial populations [40].

To evaluate specific genera that cause the difference in the semen quality, the most abundant genera were compared between low- and high-quality samples. The results indicated that genera belonging to the Enterobacteriaceae family were isolated by a cultural method in our study; these bacteria, such as *Proteus*, *Prevotella*, and *Escherichia–shigella*, were reported to cause undesired effects on sperm quality [2,5]. These genera tended to be higher in the low-quality samples than in the high-quality samples. However, a significant difference was detected in the *Alysiella* and *Myroides* genera, which showed a higher abundance in the low-quality samples than in the high-quality ones, and members of the Bacilli class had a higher abundance in the high-quality samples than in the low-quality ones. This suggests that specific bacterial genera, which are less abundant in the seminal microbiome, cause the difference in boar semen quality. Similarly, Baud et al. [7] also stated that the overall bacterial content of human semen might not play a major role in male infertility, yet small subsets of microbes might impact the spermatozoa physiology during sperm transition. Interestingly, we also detected significant correlations among bacteria abundance in the boar semen. For instance, the LB order was negatively correlated with EB. While some LB are beneficial for sperm quality, such as *Lactobacillus* spp., the harmful effects on sperm quality resulted from EBs such as *Proteus* spp. Specifically, in the present study, *Escherichia–shigella*, an EB, correlated negatively with LB. Moreover, in the same EB family, *Proteus* and *Escherichia–shigella* were positively correlated. 

### 4.3. Differences in the Bacteria Detected by Different Techniques

In the current study, the culture of boar semen samples followed by identification with MALDI-TOF revealed that at least 20 different bacterial genera were present. Most identified bacteria belonged to the families Enterobacteriaceae, Pseudomonadaceae, and Pasteurellaceae. This aligns with findings from other studies [2,13,28], which have indicated that nearly all ejaculates examined contained at least 25 different bacterial types, encompassing both Gram-negative and Gram-positive bacteria, primarily from the families Enterobacteriaceae and Pseudomonadaceae. Althouse et al. [1] reported isolating 10 to 15 colonies from boar semen samples. These consistent findings provide strong evidence of bacterial contamination in boar semen during collection and subsequent laboratory processing. Moreover, the identification of a much higher number of genera, 270, using metagenomic techniques and a bioinformatics analysis, highlights the effectiveness of a bacterial culture and MALDI-TOF identification techniques in determining the extent of contamination.

The sequencing result indicated that genera belonging to the LB order (11.8%), Bacilli class (6.7%), and *Escherichia–shigella* (6.2%) were dominant candidates in the current seminal microbiome community. This result is different from that of Zhang et al. [5], who revealed that *Pseudomonas*, *Lactobacillus*, and *Ralstonia* were the dominant genera with 34.4%, 19.9%, and 6.8%, respectively. For human sperm microbiota, the dominant genera are *Lactobacillus* (19.9%), *Pseudomonas* (9.9%), and *Prevotella* (8.5%) [11]. Baud et al. [7] found that *Corynebacterium* and *Prevotella* were the most abundant bacteria in human semen. The differences in the most abundant genera across studies indicate the diversity among bacterial communities, which are not only seen among different sites of the body but even in specific microbiota, such as porcine semen. *Lactobacillus*—a Gram-positive anaerobic bacterium—is a major part of the group of lactic acid bacteria [41]. This has been reported as a normal genus in seminal bacteria communities [42] and was found to have a positive effect on sperm concentration [43]. *Lactobacillus* is not only a potential probiotic to maintain semen quality but is also useful in the interaction with the negative effects of *Pseudomonas* and *Prevotella* [11]. In the present study, although *Lactobacillus* was detected in 82.4% of samples (14 out of 17 samples), the abundance was 0.37% and it was not a dominant genus compared to other cited reports. There are two possible explanations for this phenomenon. First, our study indicated that the most abundant genera belonged to the LB order (11.8%), of which *Lactobacillus* is a member. It is due to that the *Lactobacillus*—an anaerobic bacterium—was unable to grow in the aerobically incubated condition in the present study and the difference in the database was used to construct the taxonomic profile across studies. Therefore, it is possible that the abundance of *Lactobacillus* in our study was detected at a low level, in contrast to the previous sequencing results [5,11]. Specifically, in the present study, the taxonomy was assigned to ASVs using the q2-feature-classifier, which is the classify-sklearn naive Bayes taxonomy classifier, against the Silva 138 99% OTU reference sequences [37] instead of BLAST searching the representative sequences set against the Greengenes database [44], or mapping based on the sequences obtained from the NCBI 16S ribosomal RNA sequence database and NCBI nucleotide collection database [45]. In a study similar to ours, using the naive Bayesian classifier to construct the taxonomic profiles at the genus level, the highest abundance was assigned to unclassified bacteria [46]. Second, the difference and diversity in dominant bacteria among seminal communities might be caused by the differences related to exposure of boars to microbes in the surrounding environment and feeding conditions, which did not contain a large number of probiotics of a *Lactobacillus* origin. This possibility is in line with Zhang et al. [5], who surmised that the diverse origins in feeding environments and conditions, microbial exposure, as well as antibiotic and probiotic treatments might lead to the diverse dominant bacteria in boar semen. Weng et al. [11] also revealed that these differences might be due to the sperm quality groups included in the experimental design and bioinformatics methods used. Mulder et al. [16] stated that different raising environments and different microbial exposures were associated with intestinal microbial diversity in pigs. 

### 4.4. Association between Bacteria and Boar Semen Qualities

The present study indicated that total bacterial count (CFU/mL) registered at 3.9–4.4 log_10_ was not the factor causing the difference in semen characteristics, e.g., sperm motility, sperm viability, acrosome integrity, membrane integrity, and mitochondrial activity. This finding is in line with Pinart et al. [47], who found that the threshold values for mesophilic aerobic bacteria could be between 10^3^ and 10^7^ CFU/mL before adverse effects on sperm quality occurred. However, the positive and negative effects of bacteriospermia on sperm quality were clearer in the present study from examining the small subset bacteria identified in the culture and semen characteristics. For example, the positive correlations between *Delftia acidovorans* and semen characteristics such as sperm motility, sperm viability, and acrosome integrity indicate a functional effect of positive bacteria on counteracting the harmful effects on pH, sperm motility, sperm viability, acrosome integrity, membrane integrity, and mitochondrial activity resulting from the contamination of other bacteria such as *Globicatella sanguinis*, *Staphylococcus* spp., *Escherichia coli*, *Proteus* spp., *Micrococcus* spp., and *Corynebacterium* spp. *Delftia acidovorans* secretes substances that inhibit the growth of *Staphylococcus epidermidis* through the production of reactive oxygen species (ROS) triggered by the tricarboxylic acid cycle (TCA) [48]. In the present study, the negative effects of *Globicatella sanguinis* on sperm characteristics such as acrosome integrity, sperm membrane functionality, and sperm mitochondrial activity were detected. Boar sperm acrosome integrity is relatively high and still used as an important parameter to indicate the fertilizing capacity of preserved boar semen [49]. In boar sperm, during the acrosomal reaction process, the sperm plasma membrane fuses with the outer acrosomal membrane and results in the release of the acrosomal contents [49]. As a result, the harmful effects of specific bacteria on fertilizing capacity should be taken into consideration. Previous studies indicate that parameters such as acrosome integrity, agglutination, osmotic resistance, and pH may be affected by certain bacteria at sufficient concentrations [14,15]. For instance, *Staphylococcus* spp. caused a moderate reduction in pH [14]. Moreover, *Pseudomonas aeruginosa* and *Clostridium perfringens* affected sperm motility and viability consistently as bacterial concentrations increased [14,15]. Similarly, Santos et al. [50] found that *Staphylococcus* spp. and *Corynebacterium* spp. were the dominant genera isolated in both semen and the foreskin mucosa of collared peccaries. Additionally, *Corynebacterium* spp. was found to negatively correlate with sperm membrane functionality and curvilinear velocity.

In the present study, *Staphylococcus* spp. (21.5%) was dominant in the culture, followed by *Micrococcus* spp. (13.6%) and *Globicatella sanguinis* (11.0%). This is different from the sequencing results. However, these isolated bacteria are also present in the seminal microbiota. This indicates that the bacterial culture outcome is a subset of the microbiota and raises questions about the cause of the differences in semen quality. It is also clear that the bacterial communities in low- and high-quality ejaculates isolated using the bacterial culture and next-generation sequencing are different. This suggests that differences in the abundance of specific bacteria and their interactions might be factors affecting sperm quality. Baud et al. [7] indicated that the total number of contaminated bacteria might not play a major role in male infertility, but microbes with low relative abundance are more harmful to sperm motility and morphology. A number of small-subset microbes in microbiota are indicated as male fertility biomarkers [9,10,11,51]. As a result, the presence and status of low-abundance microbes in semen bacterial communities is becoming a cornerstone in assessing semen quality and an intensive bacterial investigation must be performed to identify species of bacteria having detrimental effects on boar sperm quality.

## 5. Conclusions

The quality of boar semen is linked to the microbial composition within porcine semen. The prevalence of key microorganisms and the fluctuating quantities of certain minor microbial populations (such as *Globicatella sanguinis*, *Delftia acidovorans*, *Alysiella*, and *Myroides*), along with their interrelations, exert a more significant impact on sperm quality than the general bacterial community profile or the aggregate count of bacteria.

## Figures and Tables

**Figure 1 animals-13-03837-f001:**
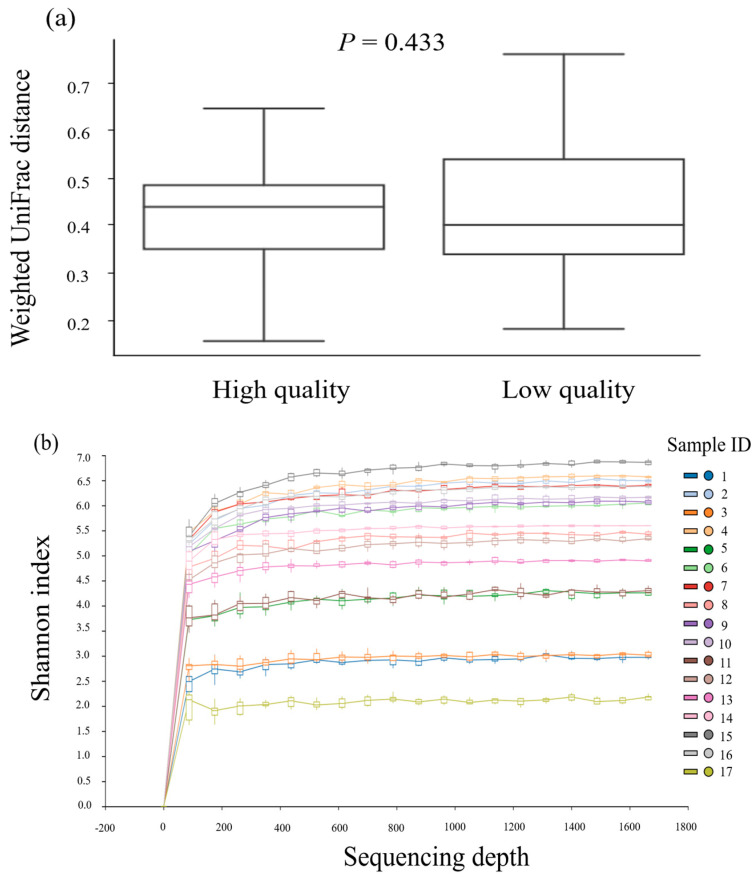
(**a**) The shifts in overall microbiota composition between low−quality and high−quality groups and (**b**) the alpha rarefaction curve across ejaculates expressed via Shannon diversity index.

**Figure 2 animals-13-03837-f002:**
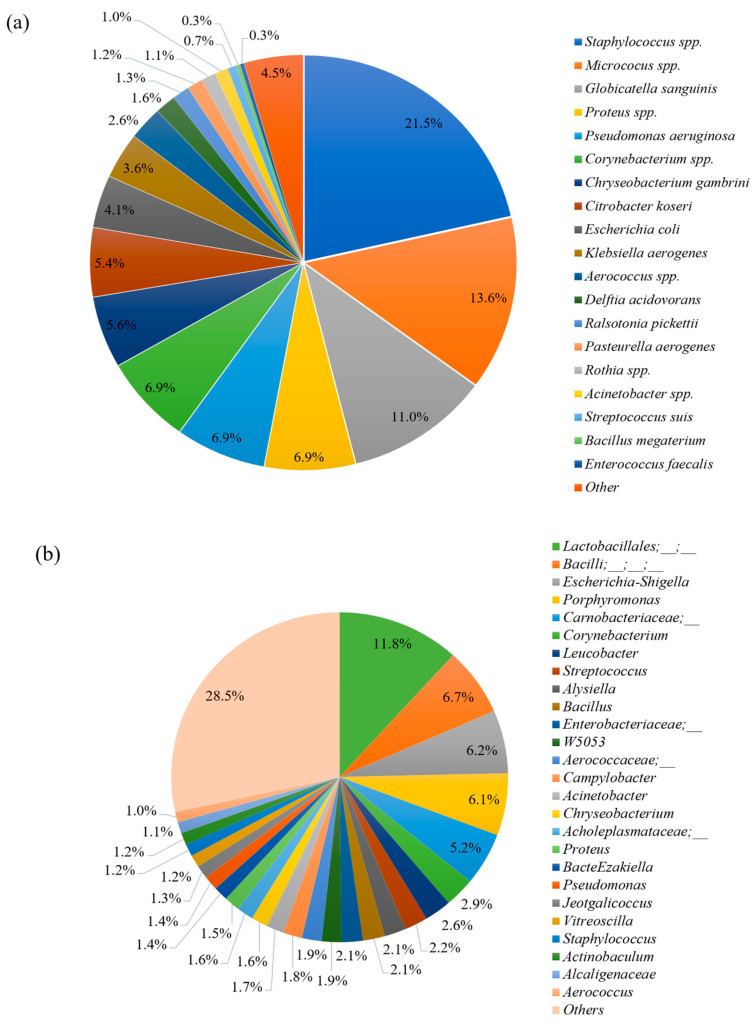
Bacterial contamination in 17 fresh boar semen samples were detected (**a**) using the bacterial culture method (**b**) using next-generation gene sequencing.

**Figure 3 animals-13-03837-f003:**
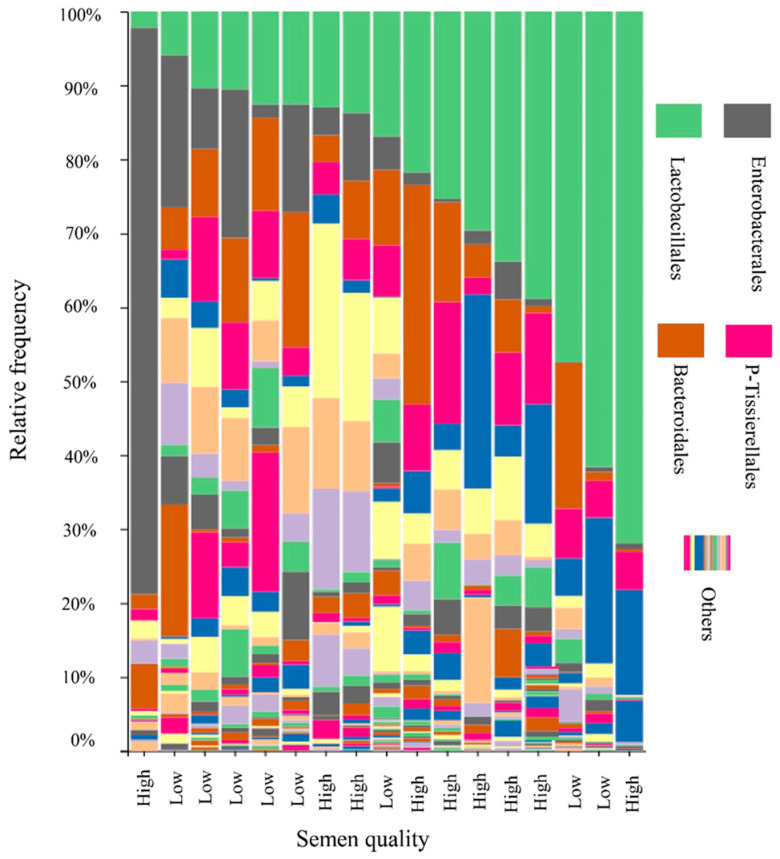
The taxonomic profiles of the bacteria communities across 17 Duroc boar semen samples detected by 16S rRNA sequencing. The samples are classified as high- or low-quality samples according to the sperm quality parameters including sperm motility and sperm viability.

**Figure 4 animals-13-03837-f004:**
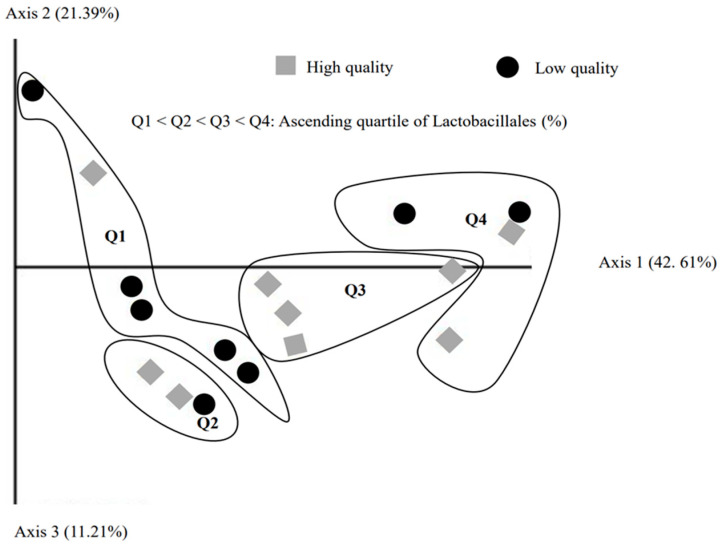
PCoA analysis of 17 semen samples was performed, classified by increasing percentage of Lactobacillales. Cluster Q1 predominantly consisted of low-quality semen samples, while clusters Q2 and Q4 contained a mixture of high- and low-quality samples. In contrast, cluster Q3 exclusively comprised high-quality semen samples.

**Figure 5 animals-13-03837-f005:**
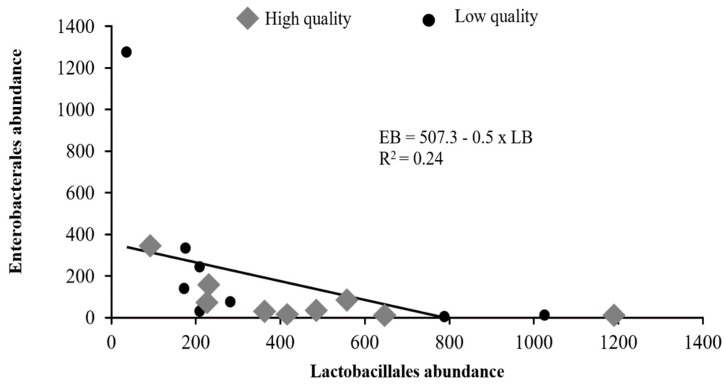
Linear relationship between dominant abundant Lactobacillales (LB) and Enterobacterales (EB) determined by gene sequencing and bioinformatics analysis in high-quality (*n =* 9) and low-quality (*n =* 8) Duroc boar semen.

**Table 1 animals-13-03837-t001:** Sperm characteristics and total bacteria count between low- and high-quality Duroc ejaculates (least-squares means ± SEM).

Variables	Group		*p* Value
	Low Quality (*n =* 8)	High Quality (*n =* 9)	
Volume (mL)	170 ± 29.8	215 ± 31.6	0.320
Concentration (×10^6^ sperm/mL)	345.0 ± 38.6	357 ± 35.9	0.321
Total sperm per ejaculate (×10^9^ sperm)	59.9 ± 9.4	54.2 ± 8.8	0.662
pH	7.6 ± 0.1	7.4 ± 0.1	0.445
Total sperm motility (%)	67.4 ± 2.8 ^a^	81.7 ± 2.7 ^b^	0.002
- Progressive motility (%)	56.0 ± 3.1 ^a^	74.6 ± 2.9 ^b^	<0.001
- Non-motile sperm (%)	32.6 ± 2.8 ^a^	18.3 ± 2.7 ^b^	0.002
- Linear velocity (VSL, µm/s)	26.6 ± 2.0	31.5 ± 1.9	0.098
- Curvilinear velocity (VCL, µm/s)	87.1 ± 4.0	94.1 ± 3.8	0.218
- Average path velocity (VAP, µm/s)	46.0 ± 2.8	51.2 ± 2.6	0.188
- Linearity (a ratio of VSL/VCL; LIN, %)	29.6 ± 2.1	35.1 ± 2.0	0.080
- Straightness (a ratio of VSL/VAP; STR, %)	53.6 ± 2.1 ^a^	60.5 ± 2.1 ^b^	0.028
- Wobbe (a ratio of VAP/VCL; WOB, %)	51.2 ± 1.8	54.4 ± 1.7	0.204
Sperm viability (%)	71.3 ± 1.3 ^a^	83.6 ± 1.2 ^b^	<0.001
Sperm acrosome integrity (%)	80.7 ± 2.0	86.8 ± 2.0	0.050
Sperm membrane functionality (%)	58.3 ± 2.4 ^a^	72.6 ± 2.2 ^b^	<0.001
Sperm mitochondrial activity (%)	67.4 ± 2.9 ^a^	79.8 ± 2.7 ^b^	0.007
Total bacterial count (CFU/mL, log_10_)	4.4 ± 0.2	3.9 ± 0.2	0.065

^a,b^ Different superscripts in each variable indicate significant differences (*p* < 0.05). Motility characteristics were measured by computer-assisted sperm analysis. Other sperm properties were evaluated by fluorescence microscopy. Total bacterial count was determined from culture on plate count agar.

**Table 2 animals-13-03837-t002:** Correlation between boar semen characteristics and both total bacterial count and the concentration of specific bacteria within the semen.

Bacterial Count (CFU/mL, log10)	Correlation Coefficient					
	pH	Motility	Viability	Acrosome Integrity	Sperm Membrane Functionality	Sperm Mitochondrial Activity
Total bacterial count	0.631 **	NS	−0.495 *	NS	−0.707 ***	NS
*Staphylococcus* spp.	0.576 *	NS	NS	NS	−0.577 *	NS
*Micrococcus* spp.	NS	NS	NS	NS	−0.529 *	NS
*Globicatella sanguinis*	NS	NS	NS	−0.699 **	−0.534 *	−0.507 *
*Pseudomonas aeruginosa*	NS	NS	NS	NS	NS	NS
*Proteus* spp.	NS	NS	NS	−0.581 *	NS	NS
*Corynebacterium* spp.	0.476 *	NS	NS	NS	NS	NS
*Chryseobacterium gambrini*	NS	NS	NS	NS	NS	NS
*Citrobacter koseri*	NS	NS	NS	NS	NS	NS
*Escherichia coli*	0.706 ***	NS	NS	NS	NS	NS
*Klebsiella aerogenes*	NS	NS	NS	NS	NS	NS
*Pasteurella aerogenes*	NS	NS	NS	NS	NS	NS
*Rothia* spp.	NS	NS	NS	NS	NS	NS
*Delftia acidovorans*	NS	0.634 ***	0.644 ***	0.500 *	0.661 **	0.512 *

NS = *p >* 0.05, * *p <* 0.05, ** *p <* 0.01, *** *p <* 0.001. Isolates from culture were identified by MALDI-TOF.

**Table 3 animals-13-03837-t003:** The most abundant bacteria (means ± SD) in low-quality Duroc boar semen samples (*n =* 8) compared with high-quality semen samples (*n =* 9).

Bacteria	Group		*p* Value
	Low Quality	High Quality	
Significant difference			
*Alysiella*	11.6 ± 10.4 ^a^	6.7 ± 10.4 ^b^	0.048
*Myroides*	7.0 ± 7.7 ^a^	10.8 ± 7.7 ^b^	0.045
*Bacilli*	6.3 ± 10.4 ^a^	11.4 ± 10.4 ^b^	0.043
Non-significant differences			
*Acholeplasmataceae*	7.3 ± 10.3	10.5 ± 10.3	NS
*Acinetobacter*	7.7 ± 10.4	10.1 ± 10.4	NS
*Actinobaculum*	8.0 ± 10.4	9.8 ± 10.4	NS
*Aerococcaceae*	9.4 ± 10.4	9.4 ± 10.4	NS
*Aerococcus*	10.5 ± 10.4	7.6± 10.4	NS
*Alcaligenaceae*	10.3 ± 10.4	7.8 ± 10.4	NS
*Bacillus*	10.5 ± 9.3	7.6 ± 9.3	NS
*Bacteroides*	8.5 ± 10.3	9.4 ± 10.3	NS
*Campylobacter*	11.0 ± 10.4	7.2 ± 10.4	NS
*Carnobacteriaceae*	6.8 ± 10.4	10.9 ± 10.4	NS
*Chryseobacterium*	9.3 ± 10.3	8.7 ± 10.3	NS
*Clostridium sensu stricto 1*	10.4 ± 10.3	7.8 ± 10.3	NS
*Corynebacterium*	10.3 ± 10.4	7.9 ± 10.4	NS
*Enterobacteriaceae*	8.7 ± 10.2	9.3 ± 10.2	NS
*Escherichia–shigella*	9.9 ± 10.3	8.2 ± 10.3	NS
*Ezakiella*	10.2 ± 10.3	7.9 ± 10.3	NS
*Flavobacterium*	10.1 ± 9.8	8.0 ± 9.8	NS
*Globicatella*	10.4 ± 10.3	7.8 ± 10.4	NS
*Jeotgalicoccus*	10.4 ± 10.4	7.7 ± 10.4	NS
*Lactobacillales*	7.6 ± 10.4	10.3 ± 10.4	NS
*Lactobacillus*	10.0 ± 10.3	8.1 ± 10.3	NS
*Leucobacter*	7.5 ± 6.9	10.3 ± 6.9	NS
*Micrococcus*	9.9 ± 10.3	8.2 ± 10.3	NS
*Mobiluncus*	11.4 ± 10.3	6.8 ± 10.4	NS
*Pasteurella*	10.3 ± 8.9	7.9 ± 8.9	NS
*Porphyromonas*	10.3 ± 10.4	7.8 ± 10.4	NS
*Prevotella*	10.1 ± 10.3	8.0 ± 10.3	NS
*Proteus*	11.2 ± 10.3	7.0 ± 10.3	NS
*Pseudomonas*	7.8 ± 10.3	10.5 ± 10.3	NS
*Rothia*	10.9 ± 10.3	7.3 ± 10. 3	NS
*Staphylococcus*	9.5 ± 10.3	8.6 ± 10.3	NS
*Streptococcus*	9.4 ± 10.4	8.6 ± 10.4	NS
*W5053*	8.9 ± 10.4	9.1 ± 10.4	NS

NS = non-significant (*p >* 0.05). ^a,b^ Different superscripts in each row indicate significant differences (*p* < 0.05). Detected from gene sequencing and bioinformatics analysis.

**Table 4 animals-13-03837-t004:** Correlation among the most abundant bacteria identified by gene sequencing and bioinformatic analysis in Duroc boar semen (*n =* 17).

Bacteria	Correlation Coefficient				
	Alysiella	Bacilli	Escherichia–Shigella	Myroides	Prophymonas
*Aerococcaceae*	NS	0.533 *	−0.506 *	−0.614 **	NS
*Bacillus*	NS	NS	NS	NS	0.705 **
*Campylobacter*	0.714 **	NS	NS	−0.486 *	NS
*Chryseobacterium*	NS	−0.498 *	0.611 **	NS	NS
*Corynebacterium*	0.723 **	NS	NS	NS	NS
*Globicatella*	NS	NS	NS	−0.503 *	NS
*Lactobacillales*	NS	0.796 ***	−0.754 ***	NS	NS
*Lactobacillus*	0.485 *	NS	NS	NS	NS
*Leucobacter*	NS	NS	NS	0.894 ***	NS
*Prevotella*	0.622 **	NS	NS	NS	0.587 *
*Proteus*	NS	−0.520 *	0.533 *	−0.495 *	NS
*Pseudomonas*	NS	NS	NS	0.704 **	NS
*Rothia*	NS	−0.740 ***	0.673 **	NS	NS
*Staphylococcus*	0.489 *	NS	NS	NS	NS
*Streptococcus*	0.547 *	NS	NS	NS	0.578 *
*Vitreoscilla*	NS	NS	NS	0.098 ***	NS
*W5053*	NS	0.484 *	−0.597 *	NS	NS

NS = *p* > 0.05, * *p* < 0.05, ** *p* < 0.01, and *** *p* < 0.001.

## Data Availability

The data presented in this study are available on request from the corresponding author. The data are not publicly available due to privacy.

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
