# Peer review of "Boar Seminal Microbiota in Relation to Sperm Quality under Tropical Environments"

_animals, 2023, doi:10.3390/ani13243837_

Round 1

Reviewer 1 Report

Comments and Suggestions for Authors

The main aim of the study was to identify the role of sperm microbiota on sperm parameters.  

It is a complex study, but there are some things that are not discussed in the paper. 

1. why did you use PBS for semen dilution? does it influence spermatic parameters? 

2. line 145 - what kind of  chamber did you use for phase contrast analysis? 

3. Line 149- Why did you not use the CASA system for determining sperm viability?

4. For molecular analysis of pathogenic microbiota why did you not isolate total bacterial flora?

5. Line 303 - what kind of technique are you referring to when you describe the results - The bacteria isolated from all ejaculates using conventional and next-generation sequencing methods - and I refer strictly to the statement - conventional?

6 Table 2 - the names of the bacteria should be in italics.  Same for Table 4. 

Author Response

# Reviewer 1

The main aim of the study was to identify the role of sperm microbiota on sperm parameters. It is a complex study, but there are some things that are not discussed in the paper.

  1. Why did you use PBS for semen dilution? does it influence spermatic parameters?

ANSWER: The reason for using phosphate buffered saline (PBS) in semen dilution is that PBS, a safe solution commonly used in biological laboratories, is effective for diluting cells and spermatozoa before microscope slide observation. Phosphate-buffered saline, with a pH of 7.2–7.4, were used to protect sperm cells from osmotic damage, such as rupturing or shrinking. Consequently, PBS can be utilized for spermatozoa dilution without negatively affecting DNA profiling or other spermatic parameters, provided the analysis is performed within 1 or 2 min. Additional explanation has been added in Materials and methods section “Phosphate-buffered saline, with a pH of 7.2–7.4, were used to protect sperm cells from osmotic damage, such as rupturing or shrinking.”. Additional reference has also been added.

References:

  1. Martin NC, Pirie AA, Ford LV, Callaghan CL, McTurk K, Lucy D, Scrimger DG. The use of phosphate buffered saline for the recovery of cells and spermatozoa from swabs. Sci Justice 2006;46(3):179-84. https://doi.org/10.1016/S1355-0306(06)71591-X.
  1. Line 145 - what kind of chamber did you use for phase contrast analysis?

ANSWER: The chamber we used for phase contrast analysis is “Hamilton 2X-CEL® Slides, Disposable Sperm Analysis Chamber, 20 micron (Hamilton Thorne Inc., Massachusetts, USA)”. This information has been added to the revised manuscript.

  1. Line 149- Why did you not use the CASA system for determining sperm viability?

ANSWER: We have used Sperm Class Analyzer (SCA® CASA system, MICROPTIC S.L., Barcelona, Spain) which has only SCA® motility and concentration module for the automatic analysis of the sperm motility and concentration in boar semen. Our machine has no SCA® vitality module to analyze sperm viability. Therefore, we did not use our CASA machine for determining sperm viability. However, our technique for determining sperm viability was using SYBR14/Ethidium homodimer-1 (Fertilight® Sperm Viability Kit, Molecular Probes Europe, Leiden, the Netherlands) which has been used in previous studies [3,23].

  1. For molecular analysis of pathogenic microbiota why did you not isolate total bacterial flora?

ANSWER: Additional explanation has been provided “Through next-generation gene sequencing, we extracted and sequenced DNA to identify total bacterial flora contaminated in the boar semen. We interpret the total isolated bacterial flora by examining (i) dominant abundant bacteria, a small subset of bacteria and their correlations, and (ii) the bacterial richness and diversity in each semen sample (alpha diversity: Chao-1, Shannon index), as well as among semen samples (beta diversity: UniFrac distance).”

  1. Line 303 - what kind of technique are you referring to when you describe the results - The bacteria isolated from all ejaculates using conventional and next-generation sequencing methods - and I refer strictly to the statement - conventional?

ANSWER: The statement “conventional” has been changed to “bacterial culture method”. The sentence has been modified as: “The bacteria detected in all ejaculates, using both bacterial culture and next-generation gene sequencing methods, are described in Figure 2. Differences re-garding dominant microbes between the two methods were observed” (L. 311-313).

  1. Table 2 - the names of the bacteria should be in italics. Same for Table 4.

ANSWER: Modified as suggested.

Reviewer 2 Report

Comments and Suggestions for Authors

Despite the importance, studies on reproductive microbiomes are relatively scarce. At this sense, the impact of the present manuscript is highlighted. It is focused on the boar seminal microbiota in relation to sperm quality under tropical environments. In general, it is well written and the hypothesis and objectives are clearly justified. Only minor issues are listed below:

1. Keywords - Try to use unedited indexing terms. Please, avoid to repeat those cited in the text.

2. In Material and methods, please provide the average age an weight of the individuals used for the experiment. Also, inform if the stud boars used for the experiment presented previously proven fertility.

- According to authors, experiment was conducted for more than one year (November 2021 to December 2022). Therefore, animals were collected during different seasons (wet and dry). If authors report in the introduction that reproductive microbiome could be influenced by season or weather, why was this not evaluated on the experiment?

3. In Results:

- In Table 1, different superscript letters would be welcome to highlight the variables in which groups presented significant differences, even if there is a column showing the p values.

- Regarding CASA analysis, didn't you evaluate LIN, STR and sperm subpopulations? Were there no differences on these parameters?

4. Discussion

- Both in discussion as in the introduction, authors highlight that boars from different breeds present distinct reproductive microbiome. Why? How a breed could interfere on reproductive microbiome? This is not more linked to the different places where experiments were conducted or even other parameters as management and environment? Please, discuss this.

- Authors should also consider to discuss their data with those from wild pigs since similar effects were found at there. Please see: Santos CS, Silva AM, Maia KM, Rodrigues GSO, Feijó FMC, Alves ND, Oliveira MF, Silva AR. Composition of semen and foreskin mucosa aerobic microbiota and its impact on sperm parameters of captive collared peccaries (Pecari tajacu). J Appl Microbiol. 2020 Sep;129(3):521-531. doi: 10.1111/jam.14663

Author Response

# Reviewer 2

Despite the importance, studies on reproductive microbiomes are relatively scarce. At this sense, the impact of the present manuscript is highlighted. It is focused on the boar seminal microbiota in relation to sperm quality under tropical environments. In general, it is well written and the hypothesis and objectives are clearly justified. Only minor issues are listed below:

  1. Keywords - Try to use unedited indexing terms. Please, avoid to repeat those cited in the text.

ANSWER: Keywords have been changed to: bacteria, bioinformatics, pig, reproduction, semen.

  1. In Material and methods, please provide the average age and weight of the individuals used for the experiment. Also, inform if the stud boars used for the experiment presented previously proven fertility.

ANSWER: Agree. Page 3, Line 103 -104 - The information of the average age and weight of the proven boar has been added: “All of the boars had previously been proven fertile and had an average age of 1.91 ± 0.46 years (ranging from 1.1 to 2.8 years) and an average body weight of 262.4 ± 37.7 kg (ranging from 193 to 342 kg).”

- According to authors, experiment was conducted for more than one year (November 2021 to December 2022). Therefore, animals were collected during different seasons (wet and dry). If authors report in the introduction that reproductive microbiome could be influenced by season or weather, why was this not evaluated on the experiment?

ANSWER: Thank you for your question. Page 3, Line 102 – 103: That is our mistake when typing the duration of semen collection. We collected all ejaculates in one month of the cool season of Thailand in 2021. We have corrected the Line 102 -103 in our revised manuscript as: “The experiment was conducted from November to December 2021”.

  1. 3. In Results:

- In Table 1, different superscript letters would be welcome to highlight the variables in which groups presented significant differences, even if there is a column showing the p values.

ANSWER: The superscript letters have been added in Tables 1 and 3.

- Regarding CASA analysis, didn't you evaluate LIN, STR and sperm subpopulations? Were there no differences on these parameters?

ANSWER: We have evaluated LIN, STR and WOB and these values have been added in the revised manuscript (Table 1). STR was different between low- and high-quality ejaculates but no differences were detected for other parameters (Table 1).

  1. Discussion

- Both in discussion as in the introduction, authors highlight that boars from different breeds present distinct reproductive microbiome. Why? How a breed could interfere on reproductive microbiome? This is not more linked to the different places where experiments were conducted or even other parameters as management and environment? Please, discuss this.

ANSWER: Thank you for your question. For the present study, we collected high and low- quality ejaculates from only Duroc boar as we have mention in Line103 of materials and methods and Line 429- 430 of discussion: “…considering all samples were from Duroc boars and collected during the cooler season”. Therefore, we did not report any data related to the differences in seminal microbiome among boar breeds in our result. However, we cited the results of Zhang et al. [5] who observed a higher alpha diversity in boar seminal bacteria in winter samples as opposed to summer ones. Additionally, in the summer, Landrace boars exhibited higher Chao1 and Shannon indices compared to Duroc and Yorkshire breeds, to introduce and discuss aspects of boar seminal microbiome. This could be explained through the differences in genetic capacity among boar breeds to adapt to changes in the environment, especially when heat stress is recognized as the main factor in summer infertility. Kraemer et al. [39] noted that during the summer, UV radiation exposure can reduce bacterial levels in boar pens. Moreover, the increased use of antibiotics as a disease prevention strategy during the summer could further contribute to a reduction in bacterial populations [40]. (L.436–444)

- Authors should also consider to discuss their data with those from wild pigs since similar effects were found at there. Please see: Santos CS, Silva AM, Maia KM, Rodrigues GSO, Feijó FMC, Alves ND, Oliveira MF, Silva AR. Composition of semen and foreskin mucosa aerobic microbiota and its impact on sperm parameters of captive collared peccaries (Pecari tajacu). J Appl Microbiol 2020 Sep;129(3):521-531. doi: 10.1111/jam.14663

ANSWER: Thank you for your suggestion. We have added additional discussion and cited the related data from this reference in our revised manuscript. (L. 546–549). “Similarly, Santos et al. [50] found that Staphylococcus spp. and Corynebacterium spp. were the dominant genera isolated in both semen and the foreskin mucosa of collared peccaries. Additionally, Corynebacterium spp. was found to negatively correlate with sperm membrane functionality and curvilinear velocity.”

Reviewer 3 Report

Comments and Suggestions for Authors

The main question of the manuscript was whether differences in the boar seminal microbiota under tropical environments can be related with boar semen quality variables, fulfilling this gap in the field. 

The present study revealed that total bacterial count was not the main factor that affects boar semen parameters. Instead of that, a positive effect on boar semen of some bacteria in low concentration (i.e., Delftia acidovorans) was noticed. This positive effect could even to overturn, the negative effects of other bacteria. This finding is new knowledge, which indicates that an intensive bacterial investigation must be performed to identify bacterial species affecting boar sperm quality, even though in low concentration in seminal plasma.

Considering the used expressions about sHOST in lines 175-176 (… to evaluate sperm plasma membrane permeability) and 183 (… indicates a functional sperm membrane), it was expected to use the same words at the presentation of the results in Table 1, page 7, but it is written Sperm membrane integrity (%). Although, it is correct, please change it to “sperm membrane functionality” following the same style of writing with “materials and methods”.

Tables and figures are the appropriate well presenting the results of the study.

Author Response

The main question of the manuscript was whether differences in the boar seminal microbiota under tropical environments can be related with boar semen quality variables, fulfilling this gap in the field. The present study revealed that total bacterial count was not the main factor that affects boar semen parameters. Instead of that, a positive effect on boar semen of some bacteria in low concentration (i.e., Delftia acidovorans) was noticed. This positive effect could even to overturn, the negative effects of other bacteria. This finding is new knowledge, which indicates that an intensive bacterial investigation must be performed to identify bacterial species affecting boar sperm quality, even though in low concentration in seminal plasma.

ANSWER: Thank you very much for reading our manuscript carefully and sharing the same interest with our research topics. We are highly appreciated and grateful for that.

Considering the used expressions about sHOST in lines 175-176 (… to evaluate sperm plasma membrane permeability) and 183 (… indicates a functional sperm membrane), it was expected to use the same words at the presentation of the results in Table 1, page 7, but it is written Sperm membrane integrity (%). Although, it is correct, please change it to “sperm membrane functionality” following the same style of writing with “materials and methods”. Tables and figures are the appropriate well presenting the results of the study.

ANSWER: Thank you for your suggestion. The phrase “sperm membrane functionality” has been used for throughout manuscript.

Once again, thank you very much for your time and effort to improve our manuscript.